# In Vitro Methodologies to Evaluate the Effects of Hair Care Products on Hair Fiber

**Robson Miranda da Gama [1,2,3,\*], André Rolim Baby [1] and Maria Valéria Robles Velasco [1]**

1    Laboratório de Cosmetologia, Faculdade de Ciências Farmacêuticas, Universidade de São Paulo,
     05508-000 São Paulo, Brazil; andrerb@usp.br (A.R.B.); mvrobles@usp.br (M.V.R.V.)
2    Laboratório de Pesquisa do Curso de Farmácia, Faculdade de Medicina do ABC,
     09060-870 Santo André, Brazil
3    LESIFAR—Laboratório Escola Semi Industrial de Farmácia, Universidade Santo Amaro,
     04829-300 São Paulo, Brazil
*    Correspondence: robson.gama@fmabc.br; Tel.: +55-11-4993-7277; Fax: +55-11-4993-5400

**Abstract:** Consumers use different hair care products to change the physical appearance of their hair, such as shampoos, conditioners, hair dye and hair straighteners. They expect cosmetics products to be available in the market to meet their needs in a broad and effective manner. Evaluating efficacy of hair care products in vitro involves the use of highly accurate equipment. This review aims to discuss in vitro methodologies used to evaluate the effects of hair care products on hair fiber, which can be assessed by various methods, such as Scanning Electron Microscopy, Transmission Electron Microscopy, Atomic Force Microscopy, Optical Coherence Tomography, Infrared Spectroscopy, Raman Spectroscopy, Protein Loss, Electrophoresis, color and brightness, thermal analysis and measuring mechanical resistance to combing and elasticity. The methodology used to test hair fibers must be selected according to the property being evaluated, such as sensory characteristics, determination of brightness, resistance to rupture, elasticity and integrity of hair strain and cortex, among others. If equipment is appropriate and accurate, reproducibility and ease of employment of the analytical methodology will be possible. Normally, the data set must be discussed in order to obtain conclusive answers to the test.

**Keywords:** hair; hair care; in vitro methodologies

## 1. Introduction

Human hair does not have a vital function, but is an important element of body image, because it has a psychological and social importance as part of one's identity. The hair is one of the few physical characteristics of the human body that can be altered according to fashion trends, culture or social values [1].

Consumers make use of different hair care products to change the physical appearance of hair, such as shampoos, conditioners, styling products, hair dye and hair straighteners [2].

A healthy, beautiful and strong hair is broadly desired. It is a fiber composed of spindle cells, made of a complex composition and contains mainly α-keratin which, depending on the type of hair, can equate to 65% to 95% of its mass [3,4].

The hair can be divided into two components: Root and shaft. Inside the follicle is the hair root, while the part which protrudes from the skin surface is called the hair shaft [3]. The hair shaft has three main structures: Cuticle, Cortex and Medulla, from outside to inside. The external layer of the hair shaft is the cuticle. Its core function is to protect the hair shaft against environmental and chemical damages. Adherence and the orientation of cuticle scale is responsible for surface properties such as

brightness and resistance to combing. The Cortex is the layer located just below the cuticle. It lends mechanical properties, such as tensile strength and elasticity to the hair fiber. In this layer, the melanin responsible for hair color is located. The inside layer (Medulla) is present only in thick hair. Its function has not been clearly defined [1–4].

Consumers are increasingly concerned about their appearance, either for aesthetic or health reasons. They expect cosmetic products to meet their needs in a broad and effective manner. Nowadays, there is great ease of access to information and this makes consumers aware and more demanding regarding the acclaimed benefits described in product labels.

Evaluating efficacy of hair care products in vitro involves use of normally highly accurate equipment. Tests are generally specific and provide information of only one hair attribute for each assay.

Those techniques have advantages over subjective evaluations: They will not demand a group of volunteers, they frequently demand only a short period of time, they use specific hair locks and must meet standards.

Properties and effects of cosmetic products on hair fiber can be assessed through many methods, such as Scanning Electron Microscopy [5], Transmission Electron Microscopy [6], Atomic Force Microscopy [7], Optical Coherence Tomography [8], Infrared Spectroscopy [9], Raman Spectroscopy [10], Protein Loss [11,12], Electrophoresis [13], color and brightness [14], Thermal Analysis [15] and measuring mechanical resistance against combing and elasticity [12,16].

## 2. Image Analysis

### 2.1. Scanning Electron Microscopy

Scanning electron microscopy (SEM) is an electron microscope capable of producing highly amplified and sharp images (up to 300,000 times from 3 to 20 nm, according to the equipment being used) of sample surfaces by scanning them with focused electron beams, under vacuum. SEM images are created by interacting electrons with the sample's atoms, resulting in various signals transformed in three-dimensional images. They are useful in order to evaluate the sample's surface topography and composition [17,18]. The SEM images, gray scale mapping, secondary electron count (SE—secondary electrons) and backscattered electrons (BSE—electron backscattering) are generated from analyzed material [18].

If the sample is not conductive and is going to be analyzed by SEM, it must first be metalized. Metallization consists of precipitation under vacuum, in a micrometer film of conductive material (e.g., gold or carbon) on the surface of the sample, enabling conduction of electrical current [18].

This could be applied to access certain hair strain properties, such as overall appearance [19]; particle deposition over surface and affinity to incorporated ingredients in hair care products [20]; structural and morphological changes [6,21,22]. This technique enables one to see samples with opalescent to opaque appearance, to amplify images and to measure composing parts [5].

### 2.2. Transmission Electron Microscopy

Transmission electron microscopy (TEM) is a technique in which a beam of electrons is transmitted through an ultra-thin sample, interacting as it passes through it. This system consists of an emitting electrons source where a set of electromagnetic lenses enclosed in a column evacuated at low pressure [23] control the beam.

When passing through the sample, the electrons interact to the lower surface with a distribution of intensity and direction controlled mainly by the diffraction laws imposed by the crystalline arrangement of the atoms in it. Then, the angular distribution of the diffracted electron beams leads to the formation of the image or the diffraction diagram for observation on the screen or on the photographic plate, with an increase in size between 1000 and 300,000 times [23].

TEM can be used to assess the ultrastructures and/or morphological changes of hair fibers after the use of cosmetic chemical treatments [6,24,25].

### 2.3. Atomic Force Microscopy

Atomic Force Microscopy (AFM) enables the acquisition of a wide range of information, because it provides many imaging possibilities for different types of samples [26].

Through this method, it is possible to study material surfaces beyond the scope of electronic and optical microscopy. Visualizing macromolecules and cell imaging with near atomic resolution is made possible. AFM has the advantage of virtually no need to make the sample conductive, as in such techniques as SEM [27].

AFM images are generated by measuring the forces of attraction or repulsion between the sample surface and a very fine needle, or probe, that scans the sample in such microscopes. This scanning is performed by means of a piezoelectric system with displacement at x, y and z coordinates with the accuracy of a tenth of an angstrom, which occurs through the applied tension range [27].

AFM could be applied to get certain hair strain properties, such as Particle deposition over surface and affinity to incorporated ingredients in hair care products [20,24], structural and morphological changes [26,28–31] as well.

### 2.4. Optical Coherence Tomography

Optical Coherence Tomography (OCT) is a new morphological method for non-invasive investigation in cosmetology. Internal sectional images of high resolution are enabled by this method for investigation of internal microstructures from living tissue [8,32].

OCT uses principles of low length coherence interferometry. Biological structure images with high resolution tomography are obtained by the reflection of photons through direct coherence by equipment [8,32].

Optical properties of the sample, such as backscattering coefficient and refractive index variation are very important to obtain OCT images. The background color in the image represents the backscattering coefficient, where white color represents the high-scattering coefficient and black color the low-scattering coefficient [32].

OCT could be utilized to evaluate structural and morphological changes on hair by hair dye [33] and hair straightening [8] treatments.

## 3. Spectroscopy Analysis

In the spectroscopic methods based on absorption, the emission of electromagnetic radiation on molecules happens when the movement of electrons between their energy levels occurs [34].

### 3.1. Infrared Spectroscopy

Infrared spectroscopy (IRS) is based on the effects of radiation absorption. IRS is a type of absorption spectroscopy in which absorbed energy belongs to the infrared region (wavelength between 700 and 50,000 nm) of the electromagnetic spectrum [34].

This methodology can be employed in compound identification or to investigate the composition of a sample, which is founded on the fact that chemical bonds of chiral centers have specific vibrational frequencies, which are equivalent to the energy levels of the molecule, also called vibrational levels. Such frequencies depend on the form of potential energy on the molecule's surface, molecular geometry, molecular weight of the atoms and optionally vibrational coupling [9,34].

Near-infrared spectroscopy can be applied in hair research when one wants to analyze physical properties, such as moisturizing action of a conditioner, structural and morphological changes on hair by bleaching and hair dye treatments [9,35].

### 3.2. Raman Spectroscopy

Raman spectroscopy is a photonic technique that can provide high resolution within a few seconds, chemical and morphological data on virtually any material, organic or inorganic compound for its identification [34,36].

If an electromagnetic wave hits the surface of a medium, a fraction of the light is reflected while the remainder is transmitted into the material. The portion of radiation transmitted through the surface is absorbed as heat and another fraction is released in the form of scattered light. The emerging light is within a small portion composed of different frequencies that occur; the process that rules this phenomenon is called Raman scattering [34].

The Raman spectrum provides a direct measurement of the energy through normal oscillation mode. In turn, it depends intrinsically on interactions between the constituent atoms. Thus, the vibrational spectrum of materials will be significantly modified in the presence of compositional and structural changes. Atomic interdiffusion quantum confinement stress effects enable Raman Spectroscopy. This is used to investigate the mechanism leading to the reduction of tensile strength of permanent waved human hair [36]; the internal structure changes in virgin human black hair keratin fibers that undergo aging [37]; the influence of chemical treatments (reduction, heating, and oxidation) on keratin fibers; and the structure of virgin white human hair resulting from a permanent hair straightening [10,38].

### 3.3. Photoluminescence Spectroscopy

In photoluminescence, a medium is illuminated, then energy is absorbed. It generates an excess of energy, an effect called photo-excitation. Photo-excitation makes the electrons of the material alter its transition to the excited state with higher energy than those of equilibrium states. When these electrons return to their equilibrium state, excess energy is ejected from the material and may include light emission (radiative process) or not (non-radiative process). Energy of the emitted radiation is related to the difference between the two electronic states involved in the transition. The amount of light emitted depends on the relative contribution to the radiative process [34,39].

When it is desired to evaluate the effects of solar radiation on the degradation of the internal structures of hair [39–42] or also hair photo-protection obtained by cosmetic ingredients [40–43], photoluminescence spectroscopy analysis can be used, because in hair amino acids susceptible to photo-degradation such as tryptophan (Trp), an intrinsic fluorophore is present. Disulphide bonds in keratin, where significant losses in Trp, prior to any increase in photo-oxidation products, are detected [42].

### 3.4. Protein Loss

The hair fiber, when exposed to physical or chemical treatments, may suffer damage to its structure and thus may be altered in the protein composition. This can be evaluated easily by a quantitative assay of the extracted hair amino acids [44,45].

Many spectrophotometric methods, over the years, have been suggested for the purpose of measuring total protein from the hair; simply, there is no single universally considered methodology in use for all media. The methods which are more used and cited in scientific literature are Lowry [11,12,44–46], Bradford [42,47] and the bicinchoninic acid (BCA) method [48].

The Lowry assay is based on the reduction of the Folin reagent by protein previously treated with copper in alkaline medium. A copper atom bonds to four residuals of amino acid. This complex reduces the Folin reagent, with the solution becoming blue with maximum absorbance at about 715 nm [11,12]. The Bradford method is based on the interaction between Coomassie brilliant blue BG-250 dye and protein macromolecules containing basic amino acids or aromatic side chains. In the pH of the reaction, the interaction between the high molecular weight protein and BG-250 dye causes the dye to balance the offset to the anionic form, for which maximum absorbance is at about 595 nm [42,47]. In the BCA

method, amino acids and polypeptide chains from the hair reduced the $Cu^{2+}$ ion to $Cu^+$ that reacts with the BCA sodium salt. The product of this reaction is a purple complex composed of $Cu^+$ and two molecules of BCA with maximum absorbance at about 562 nm [48].

### 3.5. Electrophoresis

Electrophoresis is a particle separation procedure according to its electrical charge and molecular weight. It allows the isolation of organic molecule fractions such as RNA, DNA, proteins and enzymes by its migration into a gel at the time of an electric potential application [49].

In this technique, a vat with two separated compartments is used where electrodes determine its positive and negative poles. A buffer solution that conducts electricity between the chambers is placed and the gel submerged. For the particles' migration, a voltage and electrical current is applied to the system and thus the lower molecular weight ionized molecules will be able to immigrate through the gel—as opposed to the higher molecular weight ones—and will move a greater distance, getting closer to the positive pole. The fragment distance (band) traveled from the point of application is compared to the distance that other fragments of known sizes traversed in the same gel [49].

Electrophoresis is used to evaluate the influence of chemical cosmetic treatments, light exposure and combinations of these processes in the modification of hair protein composition. Also, this process is performed with untreated hair (virgin) and after cosmetic treatment, so it is possible to assess protein amount, peptides and amino acids, due to the presence of new fractions (bands) of protein not observed in untreated hair, indicating a degradation induced by the treatment of proteins, which originally were not extractable in virgin hair [13,25].

### 3.6. Diffuse Reflectance Spectrophotometry

Diffuse reflectance spectrophotometry is based on the correlation between the energy reflectance and absorption and scattering coefficients of a sample. The color change is obtained by diffuse reflectance, is measured on a specific spectrophotometer and an integrating sphere can be used to capture the scattered light from a sample. The device scans the spectral range from 360 to 740 nm, and the diffuse light from a xenon lamp [50].

Sensing of color is subjective, and thus it is important to use analytical methods to allow discrete measurements to be carried out. The Commission Internationale L'Eclairage, CIELAB or CIE *L\* a\* b\**, developed and proposed in 1976 one model to measure color; in this model, color is divided onto three axes, broadly linear to human perception. Standard *L\**, *a\**, *b\** measurements were collected where *L\** refers to the lightness on a scale of 0 to 100, *a\** denotes the red-green color range (positive value denotes higher red values) and *b\** represents the yellow-blue color range (positive value denotes higher yellow values) [51].

The diffuse reflectance spectrophotometry was used for the measurement of luster and color of hair fiber, virgin or treated, by hair care products [14,33,43,46,50].

## 4. Thermal Analysis

Thermal analysis is defined as a "group of techniques by which a physical property of a substance and/or its reaction products is measured as a function of temperature and/or time while the substance is subjected to a controlled program of temperature and under a specific atmosphere" [52].

According to Wendlandt [53], the thermoanalytical techniques most used are thermogravimetry (TG), differential termogravimetry (DTG), differential thermal analysis (DTA), differential scanning calorimetry (DSC) and thermomechanical (TMA). TG provides information with respect to changes in mass function of time and/or under certain temperature atmosphere conditions. The experiments are done by way of a high thermo-balance sensitivity, reproducibility and rapid answer as mass changes. The obtained curves provide information on the composition and thermal stability of the sample, intermediate products and residues formed [52,53].

DSC is a thermal analysis technique where energy differences are measured according to the substance and a reference material (thermally stable), depending on temperature. The substance and reference material are subjected to a controlled program temperature. During the heating or cooling process, a sample may go through temperature changes due to exothermic or endothermic events [52,53].

For better characterization of materials, it is significant to use a combination of data from TG/DTG and DSC measurements, as the TG/DTG detects events associated with mass loss, while DSC indicates the thermal events related to mass variation [15].

Several of the authors who researched the thermal analysis applied on human hair executed TG and DSC analyses with the purpose of characterizing the keratin of human hair [54–56] and verifying the effects of hair care products on keratin structure [15,57,58].

## 5. Mechanical Resistance

Mechanical strength of the hair can be measured by suitable equipment intended for measuring strength properties or elasticity of hair fiber due to tension force or load.

Hair fibers have elastic and plastic property. Plastic property of the hair fiber is measurable when a force is applied; the hair fiber extends in part, due to this force and effect, stretching about 2% of its original length. After the elastic phase, hair begins the plastic phase when the hair stretches quickly, approximately 25% to 30% in length, with a moderate increase in load. Maintaining the value of the constant force applied, the fibers stretch proportionally to the load until rupture occurs [59,60]. The lower the value of the breaking strength, the more damaged is inflicted on the cortex [12,16].

To evaluate this feature, a device has been used called the Instron Tensile Tester commercially or Dia-Stron MTT (modified dynamometer) or texturometer. Hair strands are fixed on a support, oriented from root to tip; the device applies a force or load to disrupt the fibers, which is registered by software, converting the information into a load graph of elongation [12,14,16,61].

Straightening and bleaching alter the resistance to breakage and tensile properties of human hair [12,62], while conditioner does not interfere with the test results [14,16].

## 6. Combing Resistance Analyses

Combing can be defined as the subjective perception of difficulty or ease in which hair can be combed. This is directly related to the forces that oppose the action of combing hair. This is an important attribute in the conditioning of hair and in consumer perception; the improvement reflects the ease of combing hair [63].

Assessing combing requires a modified dynamometer or texturometer with two non-metallic combs placed on a support [12,64,65].

This method consists of hanging a hair swatch over a force-measuring device (dynamometer or texturometer), making use of a comb close to the root end of the swatch, setting the comb in a right combing action directly through the swatch at a standard speed, and recording without interruption the forces that resist its motion from the point of insertion until it clears the tip end of the swatch [64,65].

Hair straightening alters the resistance to the combing of human hair, while additionally, the conditioner agents in hair straightening improve the test results [12].

## 7. Methodology to Support the Claims of Hair Care Products

Table 1 presents a summary of the methodologies applied to test each claim of hair care products.

**Table 1.** A summary table of the methodologies applied to test each claim of hair care products.

| Claims | Methodologies Applied |
| --- | --- |
| Damage hair reduction | Protein loss [11,12,44–48]; Electrophoresis [13,25]; Scanning Electron Microscopy [5,19–22]; Transmission Electron Microscopy [6,24,25]; Atomic Force Microscopy [7,26,28–31]; Optical Coherence Tomography [8,33]; RAMAN spectroscopy [10,32–34] |
| Hair Shine-Gloss | Diffuse reflectance spectrophotometry [14,33,43,46,50] |
| Easy to comb | Combing resistance [12,64,65] |
| Strength properties | Resistance to breakage [12,14,16,33,61,62] |
| Photoprotection | Photoluminescence spectroscopy [40–43] |
| Long-lasting color | Diffuse reflectance spectrophotometry [14,33] |
| Heat protection | Differential scanning calorimetry and Thermogravimetry [15,54–58] |
| Moisturizing hair | Near Infrared spectroscopy [9,34]; Differential scanning calorimetry and Thermogravimetry [15,54–58] |

## 8. Conclusions

Hair fiber has many properties and the current methodologies are complementary since, generally, they are employed to study one single feature. Thus, the use of many procedures enables multiple variable determinations, i.e., the elucidation of hair fiber properties, allowing wider results considering the hair attributes tested by adding cosmetic ingredients.

The methodology used for testing hair fiber must be selected according to the property or characteristic being analyzed, for instance, sensorial analysis; determination of brightness; resistance to rupture; elasticity; and integrity of hair strain and cortex, among others. If the equipment is appropriate (availability, initial cost or maintenance cost, for instance) and accurate, reproducibility and ease of employment of the analytical methodology will be considered.

Usually, the data set must be discussed in order to obtain conclusive answers to the test.

**Author Contributions:** Robson Miranda da Gama wrote the paper; André Rolim Baby and Maria Valéria Robles Velasco reviewed the paper.

**Conflicts of Interest:** The authors declare no conflict of interest.

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
