# Peer review of "In Vitro Methodologies to Evaluate the Effects of Hair Care Products on Hair Fiber"

_cosmetics, doi:10.3390/cosmetics4010002_

Round 1

Reviewer 1 Report

The manuscript should be of interest to people who read the cosmetics journals. Before being ready for publication the English language will require extensive editing and correcting. 

the authors have omitted a number of methods that should be included in their review. for example no mention of transmission electron microscopy(TEM) is mentioned. TEM is important for studying ultrastructural changes to cuticle, cortical and other fibre moieties. In addition reference to protein chemical methods such as protein composition and protein damage. Methods such as quantitative protein extraction and identification through using electrophoretic methods are important for this review to be a  comprehensive contribution.

Author Response

Dear reviewers,

Attached is the paper with revised English. 

Reviewer 2 Report

The manuscript briefly present review of experimental methods used to evaluate the efficacy of hair care products. Methods such as scanning electron microscopy, atomic force microscopy, optical coherence tomography, infrared spectroscopy, Raman spectroscopy, protein loss have been presented.

The manuscript is interested but the description of the method is too brief, there is no any in depft analysis of the possibilities presented by each method. There is a lack of examples of the results obtained with use of particular methods/devices, short description and proper references would help.

The manuscript can be published but should be supplement with the information mentioned above.

Author Response

Dear reviewers,

Attached is the paper with revised English. 

best regards

Round 2

Reviewer 1 Report

The authors have significantly improved the presentation of the manuscript , particularly the written English language. Some minor points  Line 84 should read conductive, line 227 should read ease. The conclusion  first paragraph needs to be rewritten. At the moment it is not making a clear presentation. The authors need to consider methods such as transmission electron microscopy which has the ability to monitor internal structural changes at low and high resolutions. Likewise protein damage needs to be assessed by the use of electrophoretic methods which will indicate changes in molecular size caused by peptide bond breakage.

Author Response

Cover letter

The below list of modifications in the article: “In Vitro” Methodologies to Evaluate the Effects of Hair Care Products on Hair Fiber.

1.      The conclusion was rewritten.

2.      The English language has been revised.

3.      The requested methodology was added: transmission electron microscopy and electrophoresis.

Best regards,

Reviewer 2 Report

Agree with the revision made.

Author Response

(The authors gave the same response as above.)
